# Thyroid Hormone Receptor α Controls the Hind Limb Metamorphosis by Regulating Cell Proliferation and Wnt Signaling Pathways in *Xenopus* *tropicalis*

**DOI:** 10.3390/ijms23031223

**Published:** 2022-01-22

**Authors:** Yuta Tanizaki, Yuki Shibata, Hongen Zhang, Yun-Bo Shi

**Affiliations:** 1Section on Molecular Morphogenesis, Cell Regulation and Development Affinity Group, Division of Molecular and Cellular Biology, Eunice Kennedy Shriver National Institute of Child Health and Human Development (NICHD), National Institutes of Health (NIH), Bethesda, MD 20892, USA; yuta.tanizaki@nih.gov (Y.T.); shibata@nibb.ac.jp (Y.S.); 2Center for the Development of New Model Organisms, National Institute for Basic Biology, National Institute of Natural Sciences, Okazaki 444-8585, Aichi, Japan; 3Bioinformatics and Scientific Programming Core, Eunice Kennedy Shriver National Institute of Child Health and Human Development (NICHD), National Institutes of Health (NIH), Bethesda, MD 20892, USA; hzhang@mail.nih.gov

**Keywords:** thyroid hormone, *Xenopus tropicalis*, metamorphosis, RNA-sequencing, ChIP-sequencing, hind limb formation

## Abstract

Thyroid hormone (T3) receptors (TRs) mediate T3 effects on vertebrate development. We have studied *Xenopus tropicalis* metamorphosis as a model for postembryonic human development and demonstrated that TRα knockout induces precocious hind limb development. To reveal the molecular pathways regulated by TRα during limb development, we performed chromatin immunoprecipitation- and RNA-sequencing on the hind limb of premetamorphic wild type and TRα knockout tadpoles, and identified over 700 TR-bound genes upregulated by T3 treatment in wild type but not TRα knockout tadpoles. Interestingly, most of these genes were expressed at higher levels in the hind limb of premetamorphic TRα knockout tadpoles than stage-matched wild-type tadpoles, suggesting their derepression upon TRα knockout. Bioinformatic analyses revealed that these genes were highly enriched with cell cycle and Wingless/Integrated (Wnt) signaling-related genes. Furthermore, cell cycle and Wnt signaling pathways were also highly enriched among genes bound by TR in wild type but not TRα knockout hind limb. These findings suggest that direct binding of TRα to target genes related to cell cycle and Wnt pathways is important for limb development: first preventing precocious hind limb formation by repressing these pathways as unliganded TR before metamorphosis and later promoting hind limb development during metamorphosis by mediating T3 activation of these pathways.

## 1. Introduction

Thyroid hormone (T3) is essential for organ metabolism and animal development in all vertebrates, especially during postembryonic development, a period around birth in mammals when plasma T3 level reaches the peak [1,2,3,4]. During this period, many organs, including the intestine and brain, are drastically remodeled to the adult form with distinct morphology compared to the fetal organs [5]. Thus, low T3 availability in humans causes cretinism characterized by profound mental retardation, short stature, and impaired development of the neuromotor and auditory systems [6].

T3 receptors (TRs) are members of the nuclear hormone receptor superfamily. TR and 9-cis retinoic acid receptors (RXRs) form complexes and bind to T3 response elements (TREs). In the absence of T3, these complexes recruit corepressors such as nuclear receptor corepressor (N-CoR) and silencing mediator of retinoid and thyroid hormone receptor (SMRT) and reduce histone acetylation. While, in the presence of T3, TR–RXR complexes recruit coactivators such as P300 and steroid receptor coactivators (SRCs) [1,2,4,7,8,9,10,11,12,13,14,15,16] and activate target gene expression, likely in part via histone acetylation. In mammals, the two TR genes have several alternative mRNA splicing products, including TRβ1, TRβ2, and TRα1 that can bind to T3, as well as TRα2, which is incapable of binding to T3 [17], with distinct tissue distributions. TRβ1 is expressed mainly in the inner ear, retina, and liver, while TRβ2 is predominantly expressed in the hypothalamus and pituitary [18,19]. TRα1 is predominantly expressed in the intestine, bone, muscle, heart, and the central nervous system, and its expression is activated earlier than T3 synthesis during vertebrate development [20,21]. However, the roles of TRs during postembryonic development in vertebrates are largely unknown. The main reason is the difficulty of studying mammalian embryos and neonates that depend on maternal supply for survival.

We studied T3 functions during *Xenopus* metamorphosis as a model for human postembryonic development. The changes during this process, including intestinal and brain remodeling, are regulated by T3 and resemble those occurring during mammalian postembryonic development [3,4,22]. Unlike any mammalian models, *Xenopus* develops in a biphasic process (embryogenesis and subsequent metamorphosis), making it easy to manipulate without maternal influence. By using knockout technology, several research groups, including ours, have generated TRα, TRβ, and TRα and TRβ double knockout *Xenopus tropicalis* animals to analyze the function of TR during metamorphosis [22,23,24,25,26]. These studies have demonstrated various TR subtype- and organ-dependent effects of TR knockout [27]. In particular, hind limb formation appears to occur regardless of TR knockout but with distinct developmental timing and rates. The expression of TRα in the hind limb peaks around stage 52 when the hind limb begins to form, and there is little plasma T3 [28,29]. Knocking out TRα, but not TRβ, induces precocious hind limb development, indicating that TRα plays a critical role in regulating the timing and rate of limb development [22,23,24,25,26].

To understand the molecular pathways regulated by TR, particularly TRα, during limb development, it is critical to identify TR target genes and reveal their expression changes during metamorphosis. Therefore, we carried out global RNA-seq and ChIP-seq analyses on wild type and TRα knockout hind limb and uncovered TRα-regulated biological pathways controlling limb development in *Xenopus tropicalis*. Here, we report the identification of over 700 TR-bound genes upregulated by T3 treatment in wild type but not TRα knockout tadpoles in the hind limb and evidence for the involvement of cell cycle and Wnt pathways during hind limb formation in response to T3. As Wnt signaling is also a well-known key pathway for limb organogenesis in mammals, including humans [30], our findings suggest that T3 regulates the timing of hind limb formation by regulating conserved pathways during limb development.

## 2. Results

### 2.1. Direct Target Genes of T3 in Wild Type Hind Limb at the Onset of the Metamorphosis

It has been reported that hind limb development can occur even in the absence of any of the two TR genes [22], although hind limb developmental timing and rates are altered in the absence of TRα or both TR genes [23,25,31]. To reveal how limb development is regulated by TR, particularly TRα, we carried out ChIP-seq to identify direct target genes of T3 in the hind limb at stage 54, the onset of natural metamorphosis (Appendix A, Appendix A). As a result, we identified 3425 and 2495 genes for wild-type hind limb from tadpoles without and with T3 treatment, respectively, or a total of 3714 TR-bound genes (Figure 1A, Appendix A). When Gene Ontology (GO) analysis was performed on these 3714 TR-bound genes, we observed that the GO terms related to the development and cellular processes such as cell cycle were among the most significantly enriched GO terms (Figure 1B, Supplemental Appendix A). Similarly, pathway analysis also identified that pathways related to the development and cellar processes such as Wnt and Hedgehog signaling were among the most significantly enriched pathways (Figure 1C and Supplemental Appendix A). These findings are consistent with an important role of TR in regulating limb development, particularly rapid growth via cell proliferation around stage 54, the onset of metamorphosis.

### 2.2. Direct Target Genes of TRα in Hind Limb at the Onset of the Metamorphosis

Given that knockout of TRα but not TRβ has a significant effect on limb development, we were interested in identifying TRα-regulated genes in the hind limb at the onset of metamorphosis. Using ChIP-seq, we identified 1130 and 2339 genes that remained bound by TR, presumable TRβ encoded by the remaining TRβ gene, in the TRα knockout hind limb in the absence or presence of T3, respectively (Figure 2 and Supplemental Appendix A). In total, there were 2499 genes bound by TR in the TRα (-/-) hind limb, and most of these genes were common between TRα (-/-) hind limb with and without T3 treatment (Figure 2A), suggesting that most genes were bound by TR constitutively in both wild type and TRα (-/-) hind limb. Comparing TR-bound genes in the wild type and TRα (-/-) hind limb revealed 1407 genes bound by TR only in wild type, i.e., representing 37.9% (1407/3714) of all TR-bound genes in wild-type hind limb (Figure 2B), suggesting that TRα plays a critical role in binding TR target genes in the limb at the onset of metamorphosis. To determine the biological processes and signaling pathways most likely affected by TRα knockout, we carried out GO and pathway analyses on the 1407 genes bound by TR only in the wild-type hind limb. We found that GO terms related to developmental processes and cell cycle were among the most significantly enriched (Figure 2C and Supplemental Appendix A). Pathway analysis also revealed the enrichment of developmental processes and cell cycle networks, including Wnt signal pathway, similar to the findings based on all TR-bound genes in wild-type hind limb (Figure 2D and Supplemental Appendix A), suggesting that TRα is important for regulating the developmental processes and pathways by T3 to control cell proliferation and limb growth.

### 2.3. Gene Regulation by T3 in Wild Type and TRα (-/-) Hind limb

To determine the effect of TRα knockout at the gene expression level, we carried out RNA-seq analysis (Appendix A) and compared the expression of all TR-bound genes as before [32]. The heatmaps of T3-induced gene expression changes revealed that much higher fractions of the TR-bound genes were upregulated or downregulated by T3 for the wild type-specific or common TR-bound genes than for TRα (-/-)-specific TR-bound genes (Figure 3), similar to the observations in the intestine [33]. In addition, among the genes regulated by T3, TRα knockout reduced T3-regulation, i.e., lower folds of upregulation or downregulation (Figure 3).

### 2.4. TRα Knockout Leads to Derepression of T3 Response Genes and Precocious Activation of Hind Limb Development Program

To determine the molecular mechanisms underlying the precocious hind limb formation observed in TRα (-/-) tadpoles, we compared the global gene expression profiles between wild type and TRα (-/-) hind limb at stage 54 without any T3 treatment. We found that 1938 genes had higher expression in TRα (-/-) hind limb compared with wild-type hind limb (Figure 4A and Appendix A). GO analysis of these 1938 upregulated, or derepressed genes revealed significant enrichment of genes in GO terms related to cell cycle and developmental processes (Figure 4B and Appendix A). Similarly, pathway analysis showed significant enrichment of genes in biological pathways associated with cell cycle and development (Figure 4C, Appendix A). These findings suggest that unliganded TRα functions to repress GO terms or pathways associated with cell proliferation and development in the hind limb of premetamorphic tadpole to prevent precocious development before stage 54.

We next compared the expression of the genes in the hindlimb of wild type tadpoles with or without T3 treatment and found that a total of 3552 genes were upregulated and 3733 downregulated by at least two folds or more after 18 hr T3 treatment in the hindlimb of wild type tadpoles (Figure 5A and Appendix A). Similar analysis for the TRα (-/-) tadpoles showed that only 1090 upregulated genes and 1400 downregulated genes in hindlimb after T3 treatment (Figure 5A and Appendix A), indicating that TRα knockout had a broad effect on not only T3 upregulated genes but also downregulated ones.

When we compared the T3-regulated genes in the wild-type hind limb to those in the TRα knockout hind limb, we identified 2638 and 2782 genes that were up and downregulated, respectively, in the wild type but not TRα knockout hind limb (Figure 5B and Appendix A). GO analysis of the 2638 genes upregulated by T3 only in wild type, but not TRα knockout hind limb, demonstrated the enrichment of GO terms involved in cell proliferation and development (Figure 5C and Appendix A). Likewise, pathway analysis also showed the enrichment of cell cycle-related canonical pathways (Figure 5D and Appendix A). As TRα knockout slows down the limb development during metamorphosis when T3 is present (stages 54–58), these findings suggest an important role of TRα in mediating T3 signal to activate cell cycle genes to promote cell cycle progression in hind limb development during metamorphosis between stage 54 and stage 58, when limb development is essentially complete.

### 2.5. TRα Knockout Reduces the Number of TR-Bound Genes Regulated by T3 in the Hind Limb

We next compared the TR-bound genes with genes upregulated in the wild-type hind limb after T3 treatment. We found that 899 genes were both upregulated by T3 and bound by TR, representing 24% of TR-bound genes and 25% of T3 upregulated genes (Figure 6A). In addition, another 692 genes were both downregulated by T3 and bound by TR, representing 19% of TR-bound genes and 19% of T3 downregulated genes (Figure 6B). Thus, overall, about 43% of TR-bound genes were either up- or downregulated by T3. Considering that not all T3 regulated genes are direct T3 response genes and that not all TR-bound genes are regulated by T3 at a single time point of T3 treatment, the 43% overlap is very significant, suggesting that most if not all, TR-bound genes are direct T3 response genes in the wild-type hind limb. On the other hand, for the TRα (-/-) hind limb, only 182 genes were both upregulated by T3 and bound by TR, representing only 7% of TR-bound genes and 17% of T3 upregulated genes (Figure 6C). Additionally, 162 genes were both downregulated by T3 and bound by TR in TRα (-/-) hind limb, representing 6% of TR-bound genes and 12% of T3 downregulated genes (Figure 6D). In total, only 14% of TR-bound genes were either up- or downregulated by T3 in TRα (-/-) hind limb. Thus, in TRα (-/-) hind limb, the fraction of TR-bound genes (presumed to be bound by TRβ) regulated by T3 was much lower than that in the wild-type hind limb.

When we compared the 1407 genes bound by TR only wild type, or 2307 genes bound by TR in both wild type and TRα (-/-) hind limb, with the 899 genes bound by TR and upregulated by T3 in the wild-type hind limb, we found that about 24% of the TR-bound genes, in either case, were upregulated by T3 (Figure 6E,F). This number is much higher than the 7% of genes bound by TR only in TRα (-/-) hind limb that were upregulated by T3 treatment of TRα knockout tadpoles. These findings suggest that TRα plays an important role in gene regulation by T3 during limb metamorphosis.

### 2.6. Coordinate TR-Binding and T3-Regulation of Genes in Cell Cycle and Wnt/β-Catenin Signaling Pathways by TRα during Hind Limb Development

The above analyses revealed that TRα knockout affected many signaling pathways. Most significantly among them were cell cycle and Wnt/β-catenin signaling pathways, whose genes were highly enriched among those derepressed (Figure 4) or lost binding by TR (Figure 2) or regulation by T3 (Figure 5) in TRα knockout hind limb at premetamorphic stage 54. These findings suggest that the changes in these pathways caused by TRα knockout may underlie the developmental effects of TRα knockout on hind limb development. To investigate how the genes in these pathways are affected by TRα knockout, we examined TR-binding and T3-regulation of individual genes in these pathways. We found that for both the cell cycle pathway (Figure 7) and the Wnt/β-catenin signaling process (Figure 8), many genes were coordinately bound by TR and upregulated by T3 in a TRα-dependent manner. For example, most of the genes which were bound by TR in cell cycle pathway were either bound by TR only in wild type hind limb, e.g., cyclin B, or both in wild type hind limb and TRα (-/-) hind limb, e.g., cyclin D and CKD4, but their expression was upregulated by T3 only in wild type hind limb (Figure 7). Likewise, many genes in the Wnt/β-catenin pathway were either bound by TR only in wild type hind limb, e.g., Casein kinase II, or both in wild type hind limb and TRα (-/-) hind limb, e.g., FOXM1 and GSK3 beta, while their expression was upregulated by T3 only in wild type hind limb (Figure 8). Interestingly, no gene in the cell cycle pathway (Figure 7) and only a single gene in the Wnt/β-catenin pathway (Figure 8) were bound by TR and downregulated by T3 during the 18 h treatment in wild type hind limb. Thus, TRα seems critical for direct binding and coordinated upregulation of T3 response genes to activate these pathways to promote cell proliferation and limb growth during metamorphosis.

## 3. Discussion

Because of its total dependence on T3 and easy manipulability without maternal influence, *Xenopus* metamorphosis has long served as a model to study postembryonic organ development, including tissue remodeling [4,34]. For T3-inducible genes such as TRβ, TR/RXR heterodimers function as repressors in the absence of T3 and as activators in the presence of T3. This property enables TR to play a dual role during *Xenopus* development [2,9,10,11,35,36,37,38,39]. Interestingly, recent TR knockout studies have revealed that TRα or TRβ is not essential for *Xenopus* metamorphosis, including de novo limb formation. On the other hand, knocking out TRα affects the timing and rate of hind limb development. TRα knockout enables limb to develop earlier in premetamorphic tadpoles, suggesting that TRα inhibits hind limb formation by repressing T3-responsible gene expression as unliganded TR during premetamorphosis [23]. On the other hand, TRα knockout also delays hind limb formation once T3 becomes available after stage 54 during metamorphosis [40]. Our global analyses of TR binding and gene expression here have revealed the likely molecular basis underlying the effect of TRα on hind limb development during *Xenopus tropicalis* development.

### 3.1. TRα Is Critical for Both the Binding of Many Target Genes by TR and Ensuring Sufficient Levels of TR Binding at Target Genes for Their Regulation by T3 during Limb Development

The first step in gene regulation by TR is the binding of TR to target genes in chromatin. TRα is highly expressed in the hind limb by stage 54, the onset of metamorphosis, while TRβ expression in the hind limb is very low but can be activated as a direct TR target gene upon T3 treatment of premetamorphic tadpoles. Our ChIP-seq analysis showed that TRα knockout drastically reduced that number of detectable TR-bound genes, from 3714 in wild type tadpoles to 2499 in the TRα knockout tadpoles, in the hind limb of tadpoles at stage 54, indicating that TRα is important to recognize the TR target genes and consistent with the expression profiles of TRα and TRβ during limb development. While our ChIP-seq analysis does not allow quantitative comparison of the levels of TR binding at individual target genes between the wild type and knockout animals, it is likely that the levels of TR binding at the 2499 genes bound by TRβ, the only TR expressed in the TRα knockout tadpoles, were also lower at individual genes in TRα knockout hind limb compared with those in wild type hind limb, as also suggested by the qPCR analysis of independent ChIP studies (Appendix A). This is because the total level of TR in the hind limb would be lower in the knockout tadpoles compared to wild-type tadpoles. Thus, TRα can affect TR target genes at both the number of genes bound by TR and the amount of TR bound to individual genes.

Consistent with the ability of TR to function as a repressor of T3-inducible genes in the absence of T3, we found that many genes were upregulated or derepressed by TRα knockout in the premetamorphic hind limb at stage 54 when there is little T3. Furthermore, T3 regulation of TR-bound genes in the hind limb was drastically reduced in TRα knockout tadpoles, in terms of both the fraction of genes regulated by T3 and the magnitudes of regulation for individual genes (Figure 3). This was true for all three groups of TR-bound genes: those bound by TR only in wild-type tadpoles, only in TRα knockout tadpoles, or in both wild type and TRα knockout tadpoles. Interestingly, for genes bound by TR only in the wild-type animals, there was still a small fraction of genes regulated by T3 in TRα knockout tadpoles, although at reduced magnitudes. This suggests that this small fraction of genes was still bound by TRβ in TRα knockout tadpoles, although at levels of binding not detectable by ChIP-seq but sufficient for some regulation by T3. On the other hand, for genes bound by TR in both wild type and TRα knockout tadpoles, a much smaller fraction of them were regulated by T3 at reduced magnitudes in TRα knockout tadpoles. This was likely due to lower levels of TR binding to these genes in TRα knockout tadpoles since only TRβ remained in TRα knockout tadpoles. The same appeared to be true for genes bound by TR only in TRα knockout tadpoles. Thus, TRα affects gene regulation at three levels, repressing them in premetamorphic tadpoles, enabling more genes to be regulated by T3 by increasing the number of genes bound by TR in part through increasing overall levels of TRs in the hind limb, and enhancing gene regulation by T3 through increased TR binding at individual genes.

### 3.2. TRα Regulates Pathways Such as Cell Cycle and Wnt/β-Catenin Signaling to Control the Timing and Rate of Limb Development

A significant change at the early stages of limb development is rapid cell proliferation. Thus, one would expect that cell cycle pathways are essential for limb development. In addition, early studies in different animal models have shown that Hedgehog and Wnt/β-catenin signaling pathways are required for limb development [41,42]. Interestingly, our global analyses revealed that these pathways are controlled by TRα to function at different stages of limb development. First, knocking out TRα derepressed or upregulated by T3-response genes in premetamorphic limb when little T3 is present. These genes were highly enriched with genes in the pathways and GO terms related to cell proliferation, cell cycle, and Hedgehog and Wnt/β-catenin signaling. Interestingly, such pathways and GO terms were also enriched among genes whose TR-binding became undetectable by ChIP-seq in the TRα knockout hind limb, supporting a derepression mechanism by TRα knockout in premetamorphic hind limb. Since TRα knockout causes precocious limb development, our findings suggest that TRα functions to repress these pathways in premetamorphic wild-type hind limb, when there is little T3, to prevent premature limb development.

Second, the pathways and GO terms related to cell proliferation, cell cycle, and Hedgehog and Wnt/β-catenin signaling were not only enriched among genes that lost TR-binding in TRα knockout hind limb but also among genes whose regulation by T3 was abolished in TRα knockout hind limb, providing a direct link between target gene binding and regulation by TRα. Furthermore, T3 is critical for limb development after metamorphosis begins at stage 54. Thus, once metamorphosis begins, TRα appears to control these pathways by increasing the number of genes bound by TR in these pathways and enhancing their regulation by T3. This, in turn, enhances the rate of limb development.

In summary, our study is the first report to identify molecular processes regulated by TRα during hind limb development. By analyzing the expression of T3-responsive genes via RNA-seq and direct TR binding to target genes via ChIP-seq, we have provided a comprehensive set of data on global gene regulation by TR, particularly TRα, and revealed the molecular processes involved in hind limb development. Of importance is the finding that TRα plays a central role in regulating the same groups of biological pathways, particularly those related to cell proliferation, cell cycle, and Hedgehog and Wnt/β-catenin signaling, to prevent precocious limb development in premetamorphic tadpoles, and to promote limb development when T3 levels rise after the onset of metamorphosis at stage 54. Given the conservations in vertebrate development, including a critical role of T3 during postembryonic development in all vertebrates [1,2,3,4] and a key involvement of Wnt signaling in the limb organogenesis in mammals [30], our findings here suggest that further studies on anuran limb metamorphosis will not only enhance our understanding of the molecular mechanisms of limb development but also help reveal potential genes and pathways as possible targets for regenerative medicine, particularly to improve tissue repair and regeneration.

## 4. Materials and Methods

### 4.1. Animals

All *Xenopus tropicalis* experiments were approved by the Animal Use and Care Committee of Eunice Kennedy Shriver National Institute of Child Health and Human Development (NICHD), U.S. National Institutes of Health (NIH). Wild-type *Xenopus tropicalis* were purchased from NASCO (Fort Atkinson, WI, USA). TRα (-/-) *Xenopus tropicalis* were generated by crossing TRα (+/-) male and female frogs [23]. Embryos were reared in 0.1 M Marc’s modified Ringers (MMR) in a 10 cm Petri dish for one day at 25 °C and then transferred to an 800 mL beaker for three days. Four days after fertilization, embryos were transferred into a large volume (9-L) container and housed under a 15 h light/9 h dark cycle. Tadpoles were staged according to the description for *Xenopus laevis* [43].

### 4.2. Chromatin Immunoprecipitation-Sequencing (ChIP-Seq) and ChIP-PCR Analysis

Tadpoles treated with 10 nM T3 for 18 h or without T3 treatment as the control were sacrificed, and the chromatin was isolated from the hind limb of at least five tadpoles per sample as described [33,44]. The hind limbs were placed in 1 mL of nuclei extraction buffer (0.5% Triton X-100, 10 mM Tris-HCl, pH 7.5, 3 mM CaCl2, 0.25 M sucrose, with the protease inhibitor tablet (Roche Applied Science, Complete, Mini, EDTA-free), 0.1 mM dithiothreitol in Dounce homogenizers on ice and crushed with 20–25 strokes by using pestle A (DWK Life Sciences (Kimble)). The homogenate was fixed in 1% formaldehyde with rotation at room temperature for 20 min before stopping the fixation with 0.1 M Tris-HCl, pH 9.5. The homogenate was then centrifuged at 2000× *g* at 4 °C for 2 min, and the pellet was resuspended in 1 mL of nuclei extraction buffer and re-homogenized in Dounce homogenizers with 10–15 strokes using pestle B. The homogenate was filtered through a Falcon 70-μm cell strainer and centrifuged at 2000× *g* at 4 °C for 2 min. The resulting pellet was resuspended in 200 μL of SDS lysis buffer (Merck Millipore Bioscience, Billerica, MA, USA) on ice, sonicated using the Bioruptor UCD-200 (Diagenode, Sparta, Greece). The output selector switch was set on High (H), and sonication was 1 h. The samples were next centrifuged at 16,000× *g* for 10 min at 4 °C. The chromatin in the supernatant was quantitated. The chromatin was adjusted to 100 ng DNA/μL by using the SDS lysis buffer and frozen in aliquots at −80 °C. Before analysis, chromatin DNA was diluted to 10 ng/μL with ChIP dilution buffer (Merck Millipore Bioscience). After preclearing with salmon sperm DNA/protein A-agarose (Merck Millipore Bioscience), input samples were taken, and 500 μL of each chromatin sample was added to a 1.5 mL tube with anti-TR antibody or anti-ID14 control antibody [45] and salmon sperm DNA/protein A-agarose beads. The mixture was incubated with rotation for 4 h at 4 °C. After incubation, chromatin immunoprecipitation assay was performed by using a ChIP Assay Kit (Merck Millipore Bioscience) according to the manufacturer’s instruction. The ChIP DNA was purified using the Nucleospin^®^ DNA extraction kit (Macherey–Nagel, Duren, Germany) and eluted with 40 μL of TE buffer. Then, the ChIP DNA was analyzed by qPCR with a TaqMan probe to determine the presence of the TRβ TRE region to confirm the sample quality. For high-throughput ChIP-seq, libraries were prepared from the immunoprecipitated DNA samples by DNA SMART™ ChIP-seq kit (Clontech/Takara Bio Co., Palo Alto, CA, USA). The constructed ChIP-seq libraries were sequenced on the Illumina HiSeq 2500 platform in the Molecular Genomics Core, NICHD, and three technical replicates from each sample were analyzed.

### 4.3. ChIP-Seq Data Processing

Raw sequencing data in FASTQ format were aligned to *X. tropicalis* genome assemblies (Xenbase v9.1) with Bowtie2 software (version 2-2.3.4.1), and redundant reads were removed from final bam files with Samtools software (version 1.9). Peak enrichments were detected with MACS2 software (version 2.1.1.20160309). The q value cutoff of 0.05 for each single bam file without control and said peaks was mapped to genes in *Xenopus_tropicalis*.JGI_4.2.90.gff3 annotation with customer R scripts. The raw read datasets for all ChIP-seq samples are available under Gene Expression Omnibus (GEO) accession number GSE193363.

### 4.4. Quantitative Real-Time PCR

Total RNA was extracted using RNeasy Plus Mini Kit (Qiagen, Valencia, CA, USA) from the hind limb of wild type and TRα (-/-) tadpoles treated with or without T3 for 18 h. Reverse transcription was carried out as described before [32]. The real-time quantitative RT-PCR (qRT-PCR) was performed in triplicates with the SYBR Green PCR MasterMix (Applied Biosystems, Foster City, CA, USA) on the Step One Plus Real-Time PCR System (Applied Biosystems) with gene-specific primers as reported [32]. The ribosomal protein L8 gene (rpl8) was analyzed as a control for normalization [46], and the gene expression analysis was performed at least twice, with similar results.

### 4.5. RNA-Sequencing (RNA-Seq) Analysis

Total RNA was extracted from the hind limb of wild type and TRα (-/-) tadpoles with or without T3 treatment for 18 h as described above. After mRNA purification using poly-T oligo-attached magnetic beads and chemically fragmentation, three cDNA libraries were generated from the same sample using the TruSeq RNA Sample Preparation Kit (Illumina, San Diego, CA, USA) as described [32]. Then, the libraries were sequenced on the Illumina HiSeq 2000 platform to obtain 100 nt paired-end reads in the Molecular Genomics Core, NICHD. The demultiplexed and adapter-removed short reads were mapped to Ensembl *Xenopus tropicalis* Genome (JGI 4.2) with STAR software (version 2.6.1c) and reads counts for each gene/exon were obtained with featureCounts tool of Subread software (version 1.6.3). R Bioconductor DESeq2 package [29] was used for gene differential expression analysis. The raw read datasets for all RNA-seq samples are available under GEO accession number GSE193364.

### 4.6. GO and Pathway Analysis

To study the potential biological significance of the changes observed in the RNA-seq and ChIP-seq, we performed pathway and gene ontological analysis by MetaCore software (GeneGo Inc., Encinitas, CA, USA). Using the gene symbol of detected genes in RNA-seq and ChIP-seq, we uploaded the list of genes in human gene names to the MetaCore software. Then we performed GO and pathway analysis by “Pathway Maps”, “Map Folders”, and “GO Processes” in the One-click Analysis tab or “Compare Experiments” in the Workflows&Reports tab.

### 4.7. Statistical Analysis

The results were analyzed using the 4-Step Excel Statistics software (OMS Publishing Inc., Tokorozawa, Saitama, Japan) and Prism 8 statistics software (GraphPad Software Inc., San Diego, CA, USA).

## Figures and Tables

**Figure 1 ijms-23-01223-f001:**
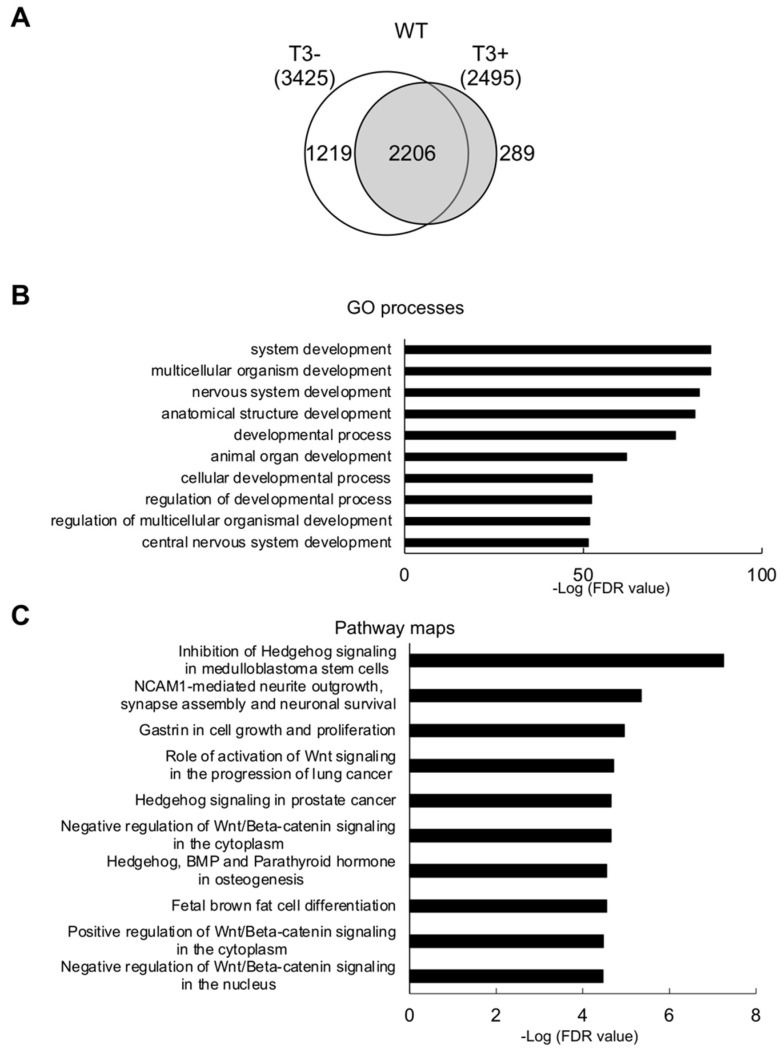
GO terms and biological pathways related to development, Hedgehog, or Wnt signaling are enriched among TR-bound genes in the hind limb. (**A**) Venn diagrams comparing TR-bound genes identified by ChIP-seq in the hind limb of premetamorphic wild-type tadpoles without and with T3 treatment reveal 2206 genes constitutively bound by TR. (**B**) Top 10 enriched GO terms related to development. The GO terms enriched among genes bound by TR in wild-type hind limb were ranked based on FDR value. The most significant GO terms related to development are shown here. (**C**) Top 10 enriched pathways related to cell cycle and development. The biological pathways enriched among genes bound by TR in wild-type hind limb were ranked based on FDR value. Note that most of the pathways are related to Hedgehog or Wnt signaling.

**Figure 2 ijms-23-01223-f002:**
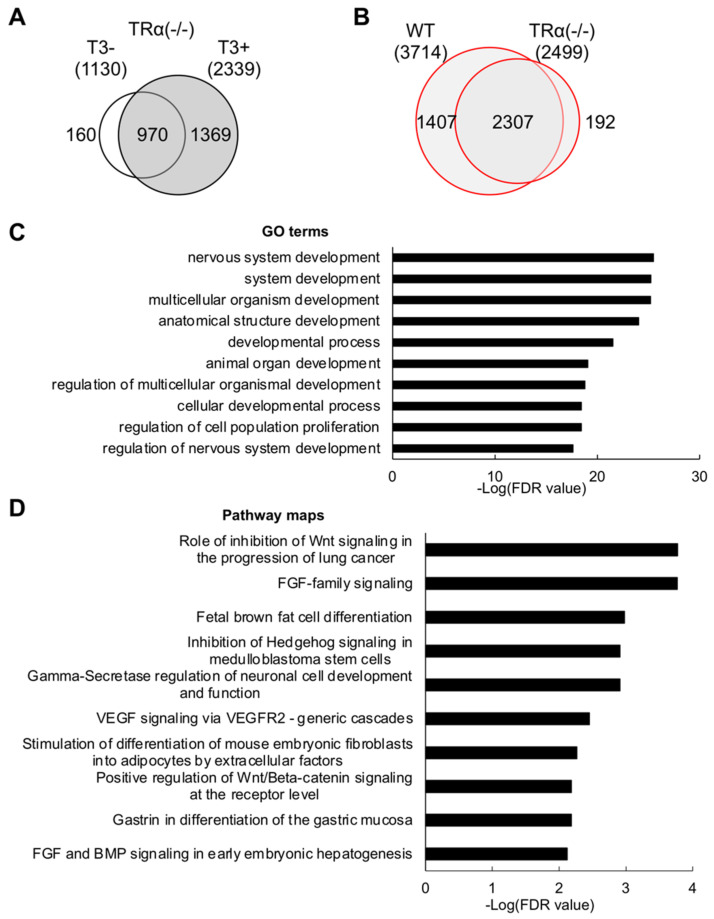
GO terms and biological pathways related to development, Hedgehog, or Wnt signaling are enriched among TR target genes bound by TR in wild type but not in TRα knockout hind limb. (**A**) Venn diagram of TR target genes identified by ChIP-seq in TRα (-/-) hind limb without and with T3 treatment. Note that, as in wild-type hind limb (Figure 1A), most genes were bound by TR constitutively. (**B**) Venn diagram comparison of all genes detected by ChIP-seq in wild type (WT) and TRα (-/-) hind limb. Of the 3714 TR target genes in wild-type hind limb, nearly 62% or 2307 genes were bound by TR in TRα (-/-) hind limb, presumably by TRβ. (**C**) GO analyses were performed using MetaCore software on the 1407 genes bound by TR in wild type but not TRα (-/-) hind limb. The top 10 most significant GO terms related to cell cycle and development were plotted here. (**D**) The pathways enriched among the 1407 genes bound by TR in wild type but not TRα (-/-) hind limb included those related to the development and Wnt signaling. The top 10 most significant pathways were plotted here.

**Figure 3 ijms-23-01223-f003:**
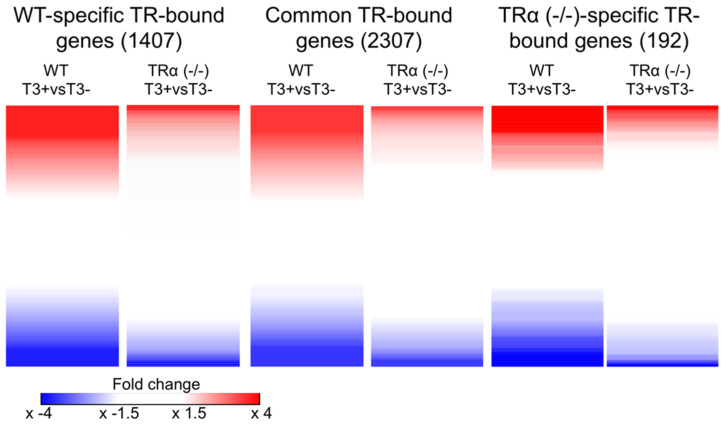
TRα knockout drastically reduces T3 regulation of TR-bound genes. Heatmap showing the fold changes by T3 treatment in the expression of TR-bound genes detected by ChIP-seq in only the wild-type hind limb (**left**), both wild type and TRα (-/-) hind limb (**middle**), or only TRα (-/-) hind limb (**right**). RNA samples were isolated from wild type and TRα (-/-) hind limb with and without 18 h T3 treatment and subjected to RNA-seq analyses. Note that a much higher fraction of the genes in each of the three classes of TR-bound genes were upregulated (**red**) or downregulated (**blue**) by T3 in the wild-type animal hind limb. In addition, TRα knockout reduced the magnitudes of T3-regulation, i.e., leading to lighter red or blue, for individual genes in the knockout hind limb, suggesting that TRα (-/-) is important for gene regulation by T3. Note that the blank regions between the red and blue areas were genes whose expression has no or little change after T3 treatment of wild type or TRα knockout animals. The color range shows fold changes, with the darkest red or blue colors showing 4-fold changes or more for the individual genes.

**Figure 4 ijms-23-01223-f004:**
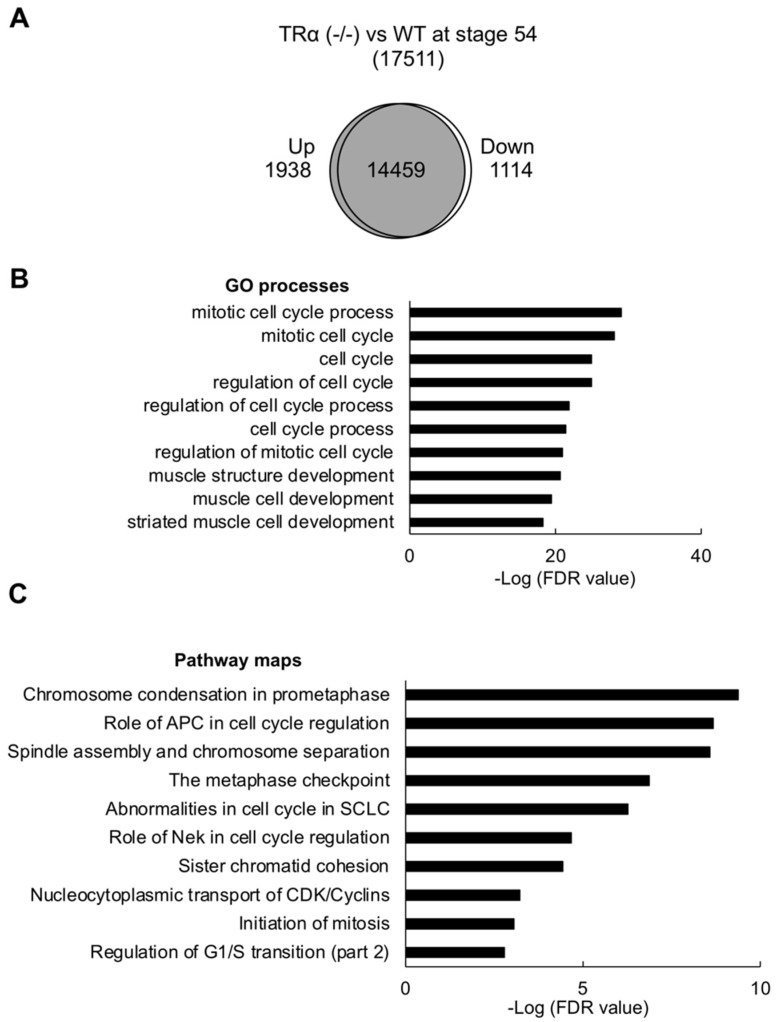
Expression of cell cycle and development-related genes are repressed by TRα in the hind limb before metamorphosis. (**A**) Venn diagram analysis for genes up or downregulated by 2-fold or more due to TRα knockout. Genes whose expression levels in the hind limbs of stage 54 tadpoles differed by 2-fold or more between wild type and TRα knockout were shown as wild type- or TRα (-/-)-specific. Note that 1938 genes were upregulated (derepressed), and 1114 genes were downregulated when comparing the expression in TRα knockout hind limb with that in the wild-type hind limb at stage 54. (**B**,**C**) Many GO terms and biological pathways related to cell cycle and development are enriched among genes upregulated (derepressed) due to knocking out TRα. GO and pathway analyses were performed on the 1938 genes upregulated (derepressed) in TRα knockout hind limb compared to the wild-type hind limb. The enriched GO terms or pathways were sorted by FDR value, and the ten most significant GO terms related to cell cycle and development (**B**) or the ten most enriched cell cycle pathways (**C**) were plotted here.

**Figure 5 ijms-23-01223-f005:**
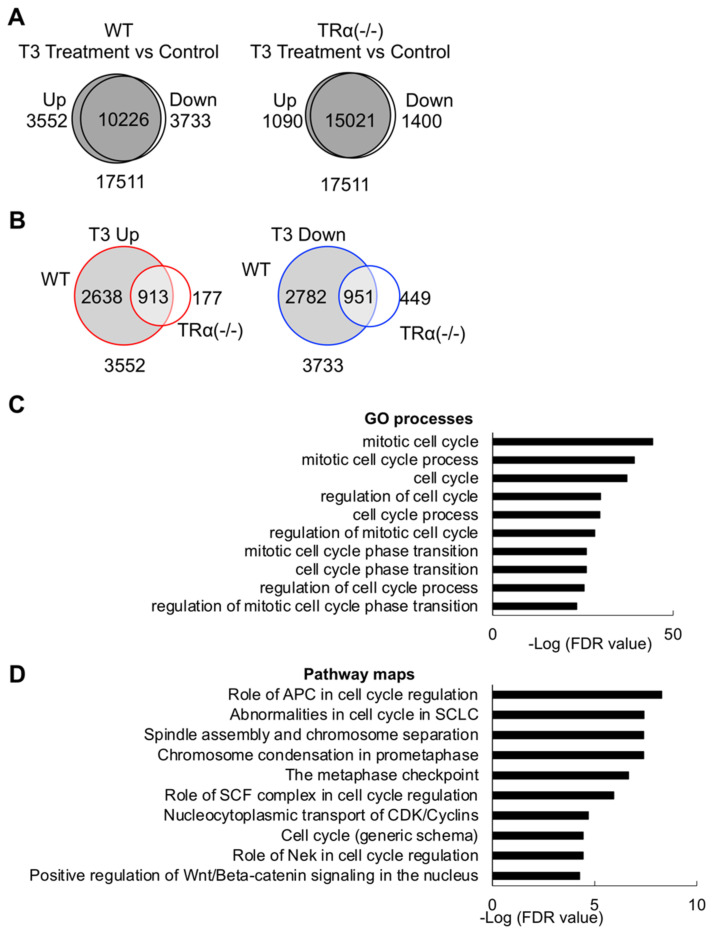
GO terms and biological pathways related to the cell cycle are highly enriched among genes whose regulation by T3 in the hind limb is abolished by TRα knockout. (**A**) Venn diagram analysis for genes upregulated or downregulated by T3 in wild type (WT, Left) and TRα (-/-) (Right) hind limb (two-fold or more and padj <0.05). Genes whose expression levels in the T3 treated vs. control hind limb differ by 2-fold or more for either wild type (WT) or TRα knockout (TRα (-/-)) tadpoles were shown as up or downregulated genes, respectively, for each genotype, while the rest of genes are shown as common. Note that many more genes were up- or downregulated by T3 in the WT compared to TRα knockout tadpoles. (**B**) Venn diagram comparison of T3 upregulated (left) or downregulated (right) genes in WT hind limb to those in TRα (-/-) hind limb reveals 2638 and 2782 genes are up- or downregulated by T3 only in WT, respectively (i.e., TRα-dependent T3 target genes). Note that most genes regulated by T3 in the TRα (-/-) hind limb were also regulated by T3 in the WT hind limb, while most genes regulated by T3 in the WT hind limb were not regulated by T3 in TRα (-/-) hind limb, indicating a major role of TRα in gene regulation by T3 in the hind limb. (**C**,**D**) Cell cycle-related GO terms (**C**) and biological pathways (**D**) are highly enriched among genes upregulated by T3 only in the WT hind limb. GO and pathway analyses were performed on the 2638 genes upregulated by T3 only in the WT hind limb. The enriched GO terms and biological pathways were sorted by FDR value. The top ten enriched cell cycle-related GO terms (**C**) and pathways (**D**) are shown here.

**Figure 6 ijms-23-01223-f006:**
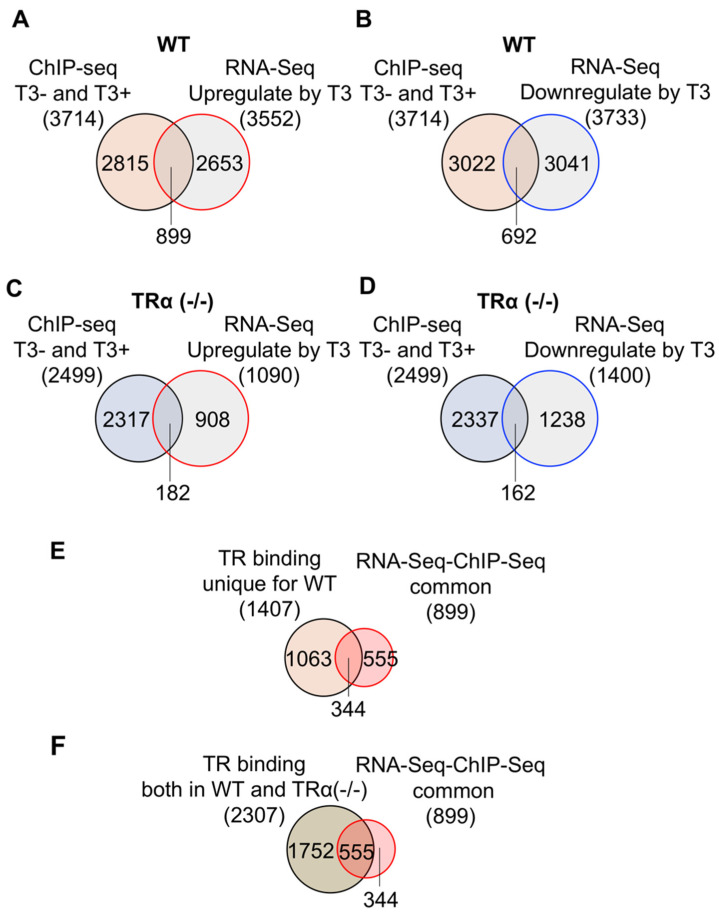
A higher fraction of TR-bound genes is regulated in the hind limb of wild-type tadpoles by T3 treatment than those in TRα (-/-) tadpoles. (**A**) Venn diagram showing the overlap between TR-bound genes identified by ChIP-Seq in WT hind limb and T3-upregulated genes in WT hind limb. Of the 3714 TR-bound genes identified by ChIP-seq, 24% were shown as the T3-upregulated genes. Conversely, 25% of the T3-upregulated genes were found to be TR-bound. (**B**) Venn diagram showing the overlap between the TR-bound genes identified by ChIP-seq in WT hind limb and T3-downregulated genes in WT hind limb. Of the 3714 TR-bound genes identified by ChIP-seq, 19% were shown as the T3-downregulated genes. Conversely, 19% of the T3-downregulated genes were found to be TR-bound. (**C**) Venn diagram showing the overlap between TR-bound genes identified by ChIP-seq in TRα (-/-) hind limb and T3-upregulated genes in the TRα (-/-) hind limb. Of the 2499 TR-bound genes identified by ChIP-seq, 7% were shown as T3-upregulated genes. Conversely, 17% of the T3-upregulated genes were found to be TR-bound. (**D**) Venn diagram showing the overlap between the TR-bound genes identified by ChIP-seq in TRα (-/-) hind limb and T3-downregulated genes in TRα (-/-) hind limb. Of the 2499 TR-bound genes identified by ChIP-seq, 7% were shown as T3-downregulated genes. Conversely, 12% of the T3-downregulated genes were found to be TR-bound. (**E**) Venn diagram comparison of the 1407 genes bound by TR only in wild-type hind limb to the 899 genes upregulated by T3 and bound by TR in the WT hind limb. Note that 24% of the genes bound by TR only in wild-type hind limb were upregulated by T3, similar to the percentage of upregulated genes among all genes bound by TR in WT hind limb (**A**). (**F**) Venn diagram comparison between 2307 genes bound by TR in both wild type and TRα (-/-) hind limb and the 899 genes upregulated by T3 and bound by TR in WT hind limb. Note that 24% of the genes bound by TR in both wild type and TRα (-/-) hind limb were upregulated by T3, similar to the percentage of upregulated genes among all genes bound by TR in WT hind limb (**A**). Thus, the data in A, E, F indicate that genes bound by TR in either wild type only or both wild type and TRα (-/-) hind limb are similarly regulated by T3 treatment of wild type premetamorphic tadpoles.

**Figure 7 ijms-23-01223-f007:**
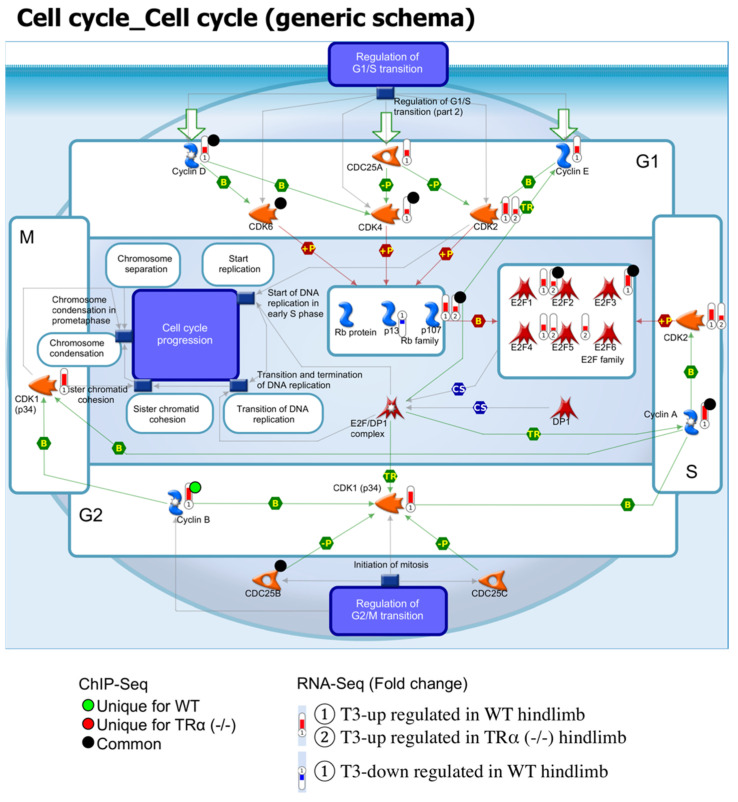
TR-binding and regulation of genes in the cell cycle pathway in hind limb based on ChIP-seq and RNA-seq data. The pathway for the generic schema of the cell cycle was visualized with regard to genes regulated by T3, based on RNA-seq, or bound by TR based on ChIP-seq. The arrows show functional interaction: green for activation. The red histograms labeled ① and ② show the genes upregulated by T3 in the hind limb of wild-type and TRα (-/-) tadpoles, respectively. The blue histogram labeled ① show the genes that were downregulated by T3 in wild type tadpoles. Green circles indicate genes bound by TR uniquely in wild type hind limb without or with T3 treatment. Red circles indicate genes bound by TR uniquely in TRα (-/-) hind limb without or with T3 treatment. The black circle indicates genes bound by TR in both wild type and TRα (-/-) hind limb. Note that most of the genes in the cell cycle pathway that were upregulated by T3 and bound by TR in wild type hind limb were not regulated by T3 in TRα (-/-) hind limb, suggesting that TRα plays an important role in mediating T3 regulation of this pathway during hind limb metamorphosis.

**Figure 8 ijms-23-01223-f008:**
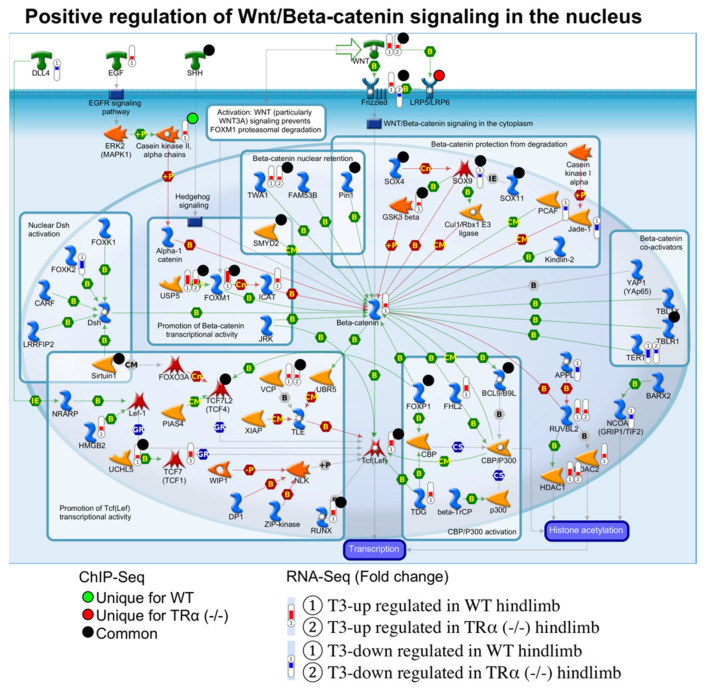
TR-binding and regulation of genes in the pathway for positive regulation of Wnt/Beta-catenin signaling in the nucleus based on ChIP-seq and RNA-seq data. The pathway for positive regulation of Wnt/Beta-catenin signaling was visualized with regard to genes regulated by T3 based on RNA-seq or bound by TR based on ChIP-seq. The arrows show functional interaction: green for activation. The red histograms labeled ① and ② show the genes upregulated by T3 in the wild-type- and TRα (-/-)- hind limb, respectively. The blue histograms labeled ① and ② show the genes that were downregulated by T3 in wild-type and TRα (-/-) hind limb, respectively. Green circles indicate genes bound by TR uniquely in wild type hind limb without or with T3 treatment. Red circles indicate genes bound by TR uniquely in TRα (-/-) hind limb without or with T3 treatment. The black circle indicates genes bound by TR in both wild type and TRα (-/-) hind limb. Note that like the cycle pathway in Figure 7, most of the genes in the Wnt signaling pathway that were upregulated by T3 and bound by TR in wild type hind limb were not regulated by T3 in TRα (-/-) hind limb, suggesting that TRα plays an important role in mediating T3 regulation of this pathway during hind limb metamorphosis.

## Data Availability

The ChIP-seq and the RNA-seq data used in this study are publicly available in the GEO database (GSE1933643 and GSE193364).

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
