# Peer review of "Thyroid Hormone Receptor α Controls the Hind Limb Metamorphosis by Regulating Cell Proliferation and Wnt Signaling Pathways in Xenopus tropicalis"

_ijms, 2022, doi:10.3390/ijms23031223_

Round 1

Reviewer 1 Report

Introduction

The text sets the scene nicely and the hypothesis of the authors is presented in detail. However, please add clearly, as the last paragraph of the section, the objectives of the study. These should be also inserted in the abstract of the manuscript.

Procedures

Xenopus tropicalis, please italicize throughout the text.

4.2. Please present a more detailed description of the technique employed as supplementary material to facilitate readers who are interested, without them searching for the original reference.

Results

The supplementary material could not be found in the submission and, hence, all the supplementary tables and figures were not evaluated.

The Results are written in a style that describes procedures. This should be corrected. All passages related to procedures (e.g., we compared, we performed etc.) should be transferred to the relevant section and the present section should contain only findings, nothing that was done. This is a serious error and must be corrected.

Discussion

This will read better if it was divided into two (or even three) sub-sections to allow readers understand the various issues more clearly.

Also, please add a passage with the clinical implications of research.

Whilst there is merit in the manuscript, the writing style, as it is now, is not supportive of acceptance at the moment. Please revise taking into account the above points and resubmit an improved version for re-evaluation.

Author Response

Introduction

The text sets the scene nicely and the hypothesis of the authors is presented in detail.

However, please add clearly, as the last paragraph of the section, the objectives of the study. These should be also inserted in the abstract of the manuscript.

Reply: Thanks for the suggestion. We have done it as suggested.

Procedures

Xenopus tropicalis, please italicize throughout the text.

Reply: Done it as suggested.

4.2. Please present a more detailed description of the technique employed as supplementary material to facilitate readers who are interested, without them searching for the original reference.

Reply: Done it as suggested.

Results

The supplementary material could not be found in the submission and, hence, all the supplementary tables and figures were not evaluated.

Reply: Sorry to hear that. We did submit them as a zip file as required.

The Results are written in a style that describes procedures. This should be corrected. All passages related to procedures (e.g., we compared, we performed etc.)should be transferred to the relevant section and the present section should contain only findings, nothing that was done. This is a serious error and must be corrected. 

Reply: We appreciate the comments. However, we feel that it is important to briefly point out the procedures/methods used to facilitate the description of the results, as commonly done for these types of papers.

Discussion

This will read better if it was divided into two (or even three) sub-sections to allow readers understand the various issues more clearly.

Reply: We appreciate the suggestion and have done so now.

Also, please add a passage with the clinical implications of research.

Reply: We thanks the reviewer for the suggestion. Our findings are very basic but understanding molecular pathways governing organ development may have implications for regenerative medicine, particularly in improving tissue repair and regeneration.  We have now briefly pointed out this in the discussion.

Reviewer 2 Report

Dear Authors,

The article sent for review is interesting, although written in a pretty complicated way and not easy to follow the authors.
This article aims to understand the mechanisms underlying vertebrates' development better, emphasizing the importance of thyroid hormone receptors.

Researchers assessed Xenopus tropicalis metamorphosis as a model of post-embryonic human development, and on this basis, demonstrated that TRα knockout induces premature development of the hind limbs.
I closely followed the study description and the methods used and found no shortcomings here. The selection of methods, the implementation of research, and then the presentation of the results prove the excellent research technique of the authors.
Figures 7 and 8 are precious, of high cognitive value, facilitating the understanding of the complex issues presented in the article. The literature is proper.

I do not have any severe criticism, but I think that the article lacks precisely formulated conclusions and answers to the question of how the obtained results can help understand the nature of things if we want to translate them into the human body mechanisms understanding.
I am a doctor, and I was looking in the article for an explanation/indication of how the results obtained by the researchers could help understand human pathologies. I did not find an answer to this question. In conclusion, I got a bit lost in the article, and I did not really understand the leading idea of ​​the researchers.
Perhaps this is a question of the nature of this article.

Hence, my question is whether the work is purely cognitive, or the results obtained in it can be used to understand the human organism's development.

Author Response

We thanks the reviewer for the comments. Given the conservations in vertebrate development, including a key role of thyroid hormone signaling, our findings here may have implications for regenerative medicine, particularly in improving tissue repair and regeneration.  We have now briefly pointed out this in the discussion.

Round 2

Reviewer 1 Report

The authors have improved the manuscript.
I am still concerned with the presentation style in the Results. I still feel that the authors mix (rather badly and inappropriately) methodological aspects of the work with their findings. This should be improved in a further revision.
The academic editor can evaluate in the re-revised manuscript the quality of changes and whether it is acceptable for publication.

Author Response

Again, we appreciate the comments. We have now revised the Result section to remove all unnecessary description related to methodological aspects.  We hope that the reviewer will find the changes are satisfactory.